# Why Petals? Naïve, but Not Experienced Bees, Preferentially Visit Flowers with Larger Visual Signals

**DOI:** 10.3390/insects14020130

**Published:** 2023-01-26

**Authors:** Nicholas J. Balfour, Francis L. W. Ratnieks

**Affiliations:** Laboratory of Apiculture & Social Insects, School of Life Sciences, University of Sussex, Brighton BN1 9QG, UK

**Keywords:** bees, behaviour, flowers, floral advertising, foraging, ray petals

## Abstract

**Simple Summary:**

Why do plants have showy flowers? Clearly, many plant species must attract pollinators and have floral adaptations for this. However, some flowers attract insects without showy petals. This suggests that a key function of showy visual signals is to attract naïve, first-time pollinator visitors. This is similar to how a restaurant with a large sign and showy visual signals may be especially important in gaining first-time visits when competing with other establishments or plants. Customers or pollinators will know if the first visit is rewarding and use this to decide whether to revisit. First, they must visit. Most flowers and restaurants would benefit from more visitors. Restaurants usually have empty tables, and many flowers are pollen limited in their reproduction. Here, we manipulated the ray petals of inflorescences of two garden flowers to test the hypothesis that the role of showy visual signals is to attract naïve visitors. On their first inflorescence visit to both species, naïve honey bees and bumble bees were more likely to visit intact inflorescences, than those with ray petals removed. However, by the tenth consecutive inflorescence, bees showed no preference. A positive correlation was observed between the visitation of inflorescences with no petals and the inflorescence number on both study plants, for both bees. These results strongly suggest that a key function of showy petals is to attract naïve, first-time visitors.

**Abstract:**

Flower evolution includes a range of questions concerning the function of showy morphological features such as petals. Despite extensive research on the role of petals in attracting pollinators, there has been little experimental testing of their importance in attracting naïve versus experienced flower-visitors. In an exploratory field study, we manipulated the ray petals of inflorescences of two garden flowers, *Rudbeckia hirta* and *Helenium autumnale,* to test the hypothesis that these showy structures primarily function to attract first-time, naïve, visitors. On their first inflorescence visit to both species, naïve honey bees and bumble bees were more likely to visit intact inflorescences, than those with ray petals removed. However, by the tenth consecutive inflorescence on the same visit to the flower patch, test insects showed no preference. A positive correlation was observed between the visitation of inflorescences with zero petals and inflorescence number on both study plants, for both bees. These results suggest that a key function of showy petals is to attract naïve, first-time visitors. Similar to how a restaurant attracts diners with a large sign, showy signals may be vital to enticing first-time visitors when competing with other establishments or plants for customers or pollinators. We hope the findings of this exploratory study will stimulate further work in this area.

## 1. Introduction

Why do plants have showy flowers? Clearly, many plant species (~85% [1]) must attract pollinators and have floral adaptations for this [2,3,4,5]. Petals or analogous flower parts are generally not green to contrast visually against background foliage [5] and ultraviolet light, a colour seen by most insects, is often part of the visual signal [6]. This suggests that a key function of showy visual signals is to attract naïve, first-time pollinator visitors. Similar to how a restaurant with a large sign, showy visual signals may be especially important in gaining first-time visits when competing with other establishments or plants. Customers or pollinators will know if the first visit is rewarding and use this to decide whether to revisit. First, they must visit. Restaurants usually have empty tables [7], and many flowers are pollen limited in their reproduction [8]. As such, most flowers and restaurants would benefit from more visitors.

Asteraceae contains over 25,000 species, are abundant on every continent [9], and are considered the largest, most successful, and highly evolved of all plant families [10]. One particular adaptation that this group possesses is the capitular inflorescence, which comprises many small flowers that open over one or more weeks [11]. In addition, many Asteraceae possess flowers with accessible floral rewards, meaning that they are generalists who cater to a wide variety of potential flower-visiting species [12,13]. The small tubular disc flowers of Asteraceae are, however, modest and, unless en masse, inconspicuous. Hence, many species have showy marginal or ray petals which are either sterile or female.

Here, we test and provide initial support for the hypothesis that showy petals are particularly attractive to first-time visitors by removing the ray petals from the inflorescences of two Asteraceae and comparing the flower choices of naïve and experienced bee visitors to control inflorescences.

## 2. Methods

### 2.1. Experimental Setup

All data were collected in a private domestic garden (Hazelmere, Magham Down, East Sussex, 50.880, 0.284) between 1000 and 1600 h during July and August 2021 and in weather conditions suitable for all flower-visitor activity (generally sunny, ≥16 °C and light wind). We used exotic Asteraceae to minimise the likelihood that flower visitors in the area had experienced these species: *Rudbeckia hirta* (var. ‘Black Eyed Susan’), and *Helenium autumnale* (var. ‘Sahin’s Early Flowerer’). These varieties were also selected for their large and conspicuous ray petals. Asteraceaeae were ideal subjects as the capitulum inflorescence is robust and easy to manipulate by removing ray-floret petals manually. Importantly, the central disc in the study species was large so the removal of petals did not render the remainder of the flower inadequate for the study insects to land on. In many flowers, the petals form part of the landing platform, so removing them would compromise insect visitation.

Each of the four study patches measured approximately 4 m^2^ and comprised 16–20 plants of a single variety in full bloom with, on average, 132 (standard deviation, ±33.6) inflorescences. The plants were in 10 l pots, 0.5–1 m in height, and were placed close together so that the distance between inflorescences was <20 cm. For the experienced flower-visitor experiment, we set out one patch of each study plant variety with inflorescences of the three treatments on 20 July 2021. For the naïve flower-visitor experiment, we set out a further patch of *H. annus* on 19 August 2021 and a further patch of *R. hirta* on 20 August 2021. 

Because honeybees and other flower-visitors show an innate preference for symmetrical flower shapes [14,15,16](Giurfa et al., 1996; Möller and Sorci, 1998; Orban et al., 2015), we used symmetrical ray petal treatments. Each capitulum was subject to one of three treatments: (i) zero ray petals (i.e., all petals removed), (ii) four ray petals in a cross arrangement, and (iii) all petals (i.e., all petals left intact). By judicially selecting which treatment newly blooming inflorescences received we ensured each patch contained equal numbers of inflorescences per treatment and that the treatments were dispersed approximately evenly across each patch. Treatments were equalised throughout the experimental period as needed. Old inflorescences were removed.

### 2.2. Inflorescence Measurements

For both study plants, we counted the number of ray petals per inflorescence and measured the diameter of ten intact inflorescences (i.e., central disc plus ray petals) and the central discs alone. These data were used to determine the relative area of visual display of the central disc or the intact flower for the two study plant species.

### 2.3. Experienced Flower-Visitors

To identify foragers that had previously visited the experimental patches (i.e., experienced visitors), from 09 to 11 August 2021, we used acrylic paint to uniquely mark actively foraging flower-visitors. During 12–14 August, flower-visitors were followed and the number and sequence of the treatment of the inflorescences visited by marked individuals were recorded. We collected data for a total of 22 h over this period.

### 2.4. Naïve Flower-Visitors

Data were collected during six study days, from 19 August to 6 September 2020. When data were not being collected plants were placed in commercially available fine mesh cages (GardenSkill 1.35 m pop-up cage, MPN: GPN100/125-04) to exclude all insect visitors after removing the cages. We patiently waited beside the patches for naïve insects to arrive and recorded the sequence in which they visited ten experimental inflorescences. To ensure we only studied naïve insects, we then captured and marked all insects with acrylic paint using a honey bee queen marking cage. Although we cannot be certain that the flower-visitors studied had not visited these flower species previously as their prior experience is unknown, they were naïve to our study patches and treatments.

### 2.5. Statistical Analysis

Statistical analyses were conducted using ‘R’ software (version 3.4.3 [17]). The assumptions of a beta regression model were met and the ‘betareg’ package was used for analysis [18]. Beta regression was used to test the relationship between the response variable, the proportion of visits to experimental treatments (zero, four or all petals), and the explanatory variable, the number of inflorescences visited. Chi-squared analyses compared raw numbers (observation proportions) to expected probabilities (equal visitation across treatments). Data visualisations were created using the package ‘ggplot2’ [19]. Bees were grouped by genus for analysis.

## 3. Results

### 3.1. Inflorescence Measurements

Both study plants had hemispherical to spherical central discs with hundreds of florets. *H. autumnale* ray petals are yellow with orange tinges. Inflorescences had 14.7 ± 2.06 (mean ± SD) ray petals. The diameter of the whole inflorescence was 64.5 ± 3.92 mm and the central disc 21.6 ± 0.70 mm. *R. hirta*’s ray florets were yellow with orange/brown tinges in some plants. Inflorescences had 13.0 ± 0.67 ray petals, with a diameter of 108.5 ± 19.51 mm, with a central disc of 21.6mm ± 0.70. Given that the total inflorescence diameters were approximately 3 and 5 times that of the central disc, the ray petals provided by far the greater area of visual display, c. 8 and 24 times that of the central disc. 

### 3.2. Experienced Visitors

We recorded ten uniquely marked experienced honey bees foraging on each of the two study flowers and followed them for an average of 23 consecutive inflorescences. These bees visited significantly fewer *R. hirta* inflorescences with no petals compared to those with four or all petals (no petals: 68, four petals: 99, all petals: 103; χ^2^ = 8.156, df = 2, *p* = 0.017). However, there was no visitation trend on *H. autumnale* inflorescences (58, 65, 60; χ^2^ = 0.426, df = 2, *p* = 0.808).

### 3.3. Naïve Visitors

We recorded 42 naïve honey bees (*R. hirta*: 23; *H. autumnale*: 19) and 28 bumble bees (*Bombus terrestris/lucorum and B. pascuorum*; *R. hirta*: 17; *H. autumnale*: 11) foraging on ten consecutive inflorescences. 

The majority of naïve honey bees (*R. hirta*: 0.83; *H. autumnale*: 0.73) and bumble bees (*R. hirta*: 0.83; *H. autumnale*: 0.72) initially visited inflorescences with all petals on both plant species, with a very low proportion visiting those with zero petals (honey bees: 0.04, 0.06; bumble bees: 0.00, 0.18). This difference was significant for honey bees on both plants (*R. hirta*: χ^2^ = 25.391, df = 2, *p* < 0.001; *H. autumnale*: χ^2^ = 13, df = 2, *p* = 0.002). However, by the tenth inflorescences, all three treatments were visited by honey bees in roughly equal proportions and did not differ significantly across the three treatments on either plant species (*R. hirta*: χ^2^ = 1.652, df = 2, *p =* 0.438; *H. autumnale*: χ^2^ = 0.333, df = 2, *p* = 0.847). Bumble bees followed a similar pattern, predominately visiting *R. hirta* inflorescences with all petals initially (χ^2^ = 21, df = 2, *p* < 0.001) and by the tenth flower, no preference was evident (χ^2^ = 1.33, df = 2, *p* = 0.513). Sample sizes were not great enough to allow the analysis of the *H. autumnale* bumble bee data. Nevertheless, the trends in the data followed a similar trajectory (Figure 1).

Overall, the data show a negative correlation between the proportion of inflorescences with all petals visited by both honey bees and bumble bees and the inflorescence number for both *R. hirta* and *H. autumnale* (Figure 1). Two of these four analyses indicated a significant correlation (Table 1). No significant trends were observed with the visitation of flowers with four petals. By contrast, a positive correlation was observed between the proportion of inflorescences with no petals visited by honey bees and bumbles bees and inflorescence number on both study plants. All four analyses indicated the correlation was statistically significant (Table 1).

## 4. Discussion

The results indicate that the additional visual signal provided by the ray petals functioned differently for naïve and experienced bee visitors. Initially, naïve honey bees and bumble bees preferentially visited treatment inflorescences with all petals on both plant species. However, very quickly and by the tenth inflorescence visit, foragers showed no preference among inflorescences with all, four, or zero ray petals.

Flowers “advertise” themselves via showy displays [20] and in doing so incur costs [21]. The conspicuous yellow ray petals of our study plants presumably served as long-distance signals that attracted naive flower-visitors to the patches. In many plant species, larger floral displays have been documented to increase the number of long-distance approaches by pollinators [22,23]. Both plant species also possess ultraviolet floral guides [6,24,25], which serve to orient bees at close range toward the reward (i.e., nectar and/or pollen), and which also known to increase visitation rates [6]. Large and colourful floral displays are known to reduce the search times and influence the flower choice of honey bees and bumble bees due to limitations in their visual resolution [26,27,28].

Numerous observational and experimental studies in other species of Asteraceae have shown that capitula with ray petals receive more visitation than rayless [4,29,30,31,32]. Our results agree with this to a degree, but with the important additional result that the ray petals primarily serve to attract naïve insects, indicating that the primary function of ray petals is to attract naïve, first-time visitors. To give a human analogy, a large showy restaurant advertisement may be attractive to first-time visitors. However, once the location and the quality of the food on offer at the restaurant are known, large signage is a less important criterion in the decision to return as the value of the resource and its location are now known. Bees are known to exhibit both long-range and short-range selectivity when foraging and the criteria can differ between the two [33]. Our results show that naïve bees very quickly changed their behaviour. This agrees with previous studies which have shown that bee foraging behaviour and flower selection is extremely flexible. For example, bees have been observed to associate display size [34], flower colour [35], and patch [36] with nectar rewards within their first three flower visits. 

Coevolution between flowering plants and pollinators has ramifications for many elements of pollinator biology. Classically, for example, there has been much emphasis on floral traits that advantage pollinators that have particular morphological features, such as recessed nectar requiring an insect with a long tongue, as in Darwin’s Orchid (*Angraecum sesquipedale*) which is pollinated by a long-tongued Sphingidae moth [37]. In the case of petals, many flowers pollinated by birds have red petals, a colour that most insects cannot see. However, in the case of the red field poppy, *Papaver rhoeas*, in its native area it and other red-petalled flowers are visited by Glaphyridae beetles that can see red. These do not occur in Europe, where following its introduction by early agriculturalists *P. rhoeas* has changed its petal colour from red to red+ultra violet, which can be seen by bees [38]. Our results also show that the ability of insects to rapidly learn may also have an important role in the evolution of floral traits. This is also seen in adaptive-colour change in floral parts, by which insects learn to avoid flowers that have changed colour to signify that they are no longer rewarding [39]. In short, the ability of bees and other pollinators to rapidly responds to differences in rewards is likely to be a significant selective force on the evolution of floral traits.

Our research provides interesting results and a novel hypothesis on the role of petals in plant-pollinator coevolution. But of course, further work on a larger scale is needed to test this hypothesis and determine the generality of our findings. For example, presenting bees with pure- and mixed-signal patches would allow investigation into the effect of long- versus. short-range signals. Moreover, further work should exclude the possibility that removing petals either reduces nectar production or releases repellent volatiles. We hope the findings of this exploratory study will stimulate further research in this area.

## Figures and Tables

**Figure 1 insects-14-00130-f001:**
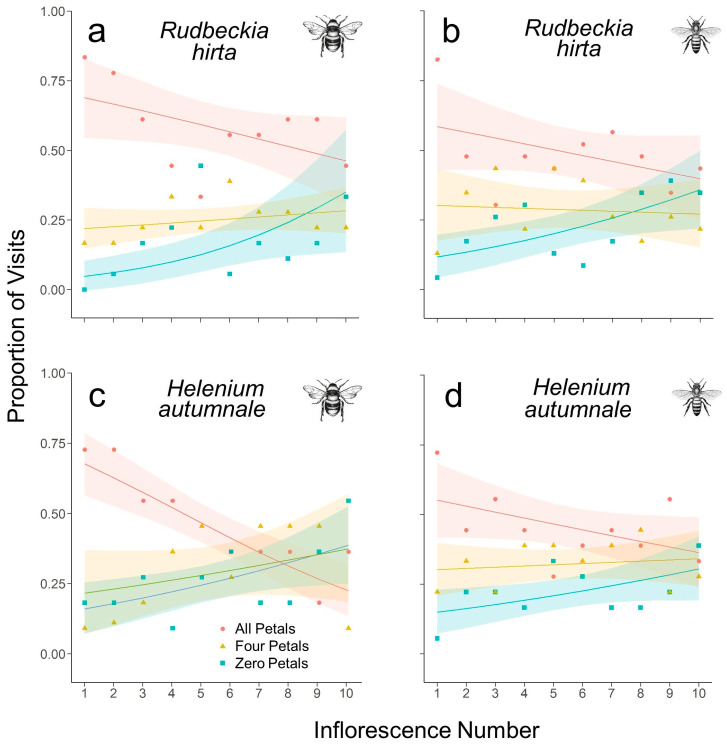
Proportion of visits by naïve bumbles bees (**a**,**c**) and honey bees (**b**,**d**) to inflorescences with all ray petals present (red circles), with four petals (yellow triangles), and those with all their ray petals removed (blue squares). The inflorescences number in the sequence of the bees’ first ten visits is given on the *x*-axis. Shown are beta regressions (lines) and 95% confidence intervals (shaded areas).

**Table 1 insects-14-00130-t001:** Output from beta regression analysis of naïve bees (Pseudo R^2^, z-value, and *p*-value) for the two study plant species, two study bee genera, and treatment types. Treatments: (i) All petals (all petals left intact), (ii) Four petals (all but four petals removed), and (ii) No petals (all petals removed).

Plant Species	Bee Genus	Treatment	R^2^	z-Value	*p*-Value
*Rudbeckia hirta*	*Apis*	All petals	0.18	−1.497	0.134
*Rudbeckia hirta*	*Apis*	Four petals	0.01	−0.309	0.757
*Rudbeckia hirta*	*Apis*	No petals	**0.40**	**2.263**	**0.008**
*Helenium autumnale*	*Apis*	All petals	**0.40**	**−2.521**	**0.012**
*Helenium autumnale*	*Apis*	Four petals	0.26	1.636	0.102
*Helenium autumnale*	*Apis*	No petals	**0.30**	**2.897**	**0.004**
*Rudbeckia hirta*	*Bombus*	All petals	0.28	−1.850	0.064
*Rudbeckia hirta*	*Bombus*	Four petals	0.13	1.067	0.286
*Rudbeckia hirta*	*Bombus*	No petals	**0.32**	**2.58**	**0.010**
*Helenium autumnale*	*Bombus*	All petals	**0.69**	**−4.782**	**<0.001**
*Helenium autumnale*	*Bombus*	Four petals	0.16	2.326	0.259
*Helenium autumnale*	*Bombus*	No petals	**0.33**	**2.386**	**0.017**

## Data Availability

Data associated with this manuscript are accessible at Figshare https://www.mdpi.com/1999-4893/16/2/112 (Balfour and Ratnieks, 2022).

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
