# Peer review of "Why Petals? Naïve, but Not Experienced Bees, Preferentially Visit Flowers with Larger Visual Signals"

_insects, 2023, doi:10.3390/insects14020130_

Round 1
Reviewer 1 Report
- Summary
- The work describes a quasi-experimental observation of real flowers in a garden. Pollinator visitations were observed at the manipulated flowers and conclusions were drawn with reference to the literature on unlearned pollinator behaviours. The methods were sound, the description is clear and I think readers would appreciate learning about this work. However, there are some issues with regards to interpretation of the results.
- The expression “naive” has been used in related literature (referenced by the authors) to refer to honeybees and bumblebees that have no prior experience with flowers. However, given the study conditions here, it’s not possible to claim that the pollinators were “naive” since their prior experience is unknown. The claim could be made that insects were new to observed flower patch.
- I am skeptical about the conclusion of the paper that this experimental set up reveals bees’ preference for large showy flowers. Given the manipulation, intact versus damaged flower petals, several alternatives are likely:
- The findings are consistent with the computational load hypothesis described in Orban et al, (2015) suggesting that radial symmetrical shapes are less costly to process than other shapes such as concentric circles. Consequently bees “innate” preference is due to the radial symmetry being “easier on the eyes”.
- I would also suggest exploring a narrative that explains the findings in the context of radial symmetry violation (Neal et. al., 1998 or Culbert & Forrest, 2016) or flower health instead of reference to naiveté or showiness.
- Excluding the confounding possibility that flowers with fewer petals contained less nectar should also be considered.
- I suggest the inclusion of figures showing appearance of flowers in the different quasi-experimental conditions
- The count of intact and manipulated flowers across the patch would be relevant to mention. Relative scarcity of manipulated flowers in relation to intact flowers would confound the results.
Author Response
Please see our responses attached.

Reviewer 2 Report
In their manuscript Balfour and Ratnieks compare bee visitation proportions between inflorescences of two Asteraceae species with all petals present, and four and all petals removed to investigate the importance of petals in attracting naïve flower visitors.
The study idea is novel and interesting. Although I agree with the scientific basis of the hypothesis, I believe the authors could have made a better job in driving the readers through the ecological-evolutional background (particularly line 48 sounds a little like in medias res). The methods used are appropriate but, in my opinion, some parts should be clearer (please see detailed comments). The presentation of the results is good. I found the Discussion slightly vague, with two full paragraphs (of the five) belonging more to the Introduction rather than there (please see detailed comments). In my opinion, this topic is interesting enough to fully put it into context and discuss its evolutionary importance. Moreover, the study is clearly on a small-scale which should be listed as a limitation.
1) Lines 14 and 15: the logical link between empty restaurant tables and pollen limited flowers is unclear.
2) Line 16: It is unclear what the “this hypothesis” refers to
3) Line 48: I cannot see how “this” (i.e., that some flowers have showy flowers, yet some attract insects without these) suggests the key function of showy flowers. Please explain the logic.
4) Lines 52-54: I cannot see how the restaurants are logically linked to pollen limitation. Please clarify.
5) Line 72: How many study patch did you have in total?
6) Lines 77-78: Please clarify throughout the ms which term refers to which treatment, i.e., zero/all petals were REMOVED or LEFT.
7) Line 93: Correct naïve in the subheading
8) Lines 94-97: Please state how many data collection hours did this net out.
9) Lines 122 and 135: Why were there differences between the number of observed inflorescences (10 vs 23)
10) Table 1: Please clarify whether “All petals” means all petals removed or left intact, and, similarly, with the “No petals”.
11) Lines 166-185: These paragraphs belong more to the Introduction, not to the Discussion.
12) Line 193: Although I am not familiar with this article this sounds rather like a spatial, not a temporal selectivity.
13) Line 195. It would be worth explaining when and how the behaviour has changed.
Author Response
Please see our responses attached.

Reviewer 3 Report
Dear Authors, I appreciate your effort to carry out a semi-field experiment, which we could describe almost as performed by citizen scientists.
It is really simple in it realization, but semplicity is a quality not a defect in science. Because its semplicity, I invite you in describing as best as possible all the steps and the tecniques adopted in realizing your observations, whitout negletting any particular.
You can find all my comments and suggestions in the pdf version.

Author Response
Please see our responses attached.

Reviewer 4 Report
The presented paper describes a neat and simple experiment aiming to show the possible role of showy flower signals for naive pollinators. My only concern is the description of Methods and in consequence the presentation of Results. The Methods are not detailed enough and unclear.
I understand the flower choice however, using such popular garden flower species does not sufficiently minimise the chance of having naive individuals in a garden set-up. Honeybees can cover large distances, and quite possible to find similar varieties in other gardens. Nevertheless, the results are still interesting.
Summarizing the Methods: 2 study patches were used, one for each species and each study patch contained all three inflorescence types. Is that right? Can you please rephrase the Methods or add a scheme/photo to make the experimental setup clear? The information is there, but it took me a few reads to understand it because it is divided between parts of the Methods.
What do you mean by equalising the treatment? Cutting down destroyed inflorescences and lowering the number of other types?
When did you set out the flower patches? There are two days mentioned in the text: 20 July and 19 August. However, in the first case, you started the observations on 9 August. The second observation was started immediately and covered the flowers with mesh between the observation. Can you describe this protocol better? Or add a graphic scheme to make it easier to understand? This part clearly needs a more straightforward description.
In the Methods section testing of experienced (marked) pollinators were mentioned, but in the Results section only ten honeybees are mentioned and their choices. This seems inconsequent. What about bumblebees?
Author Response
Please see our responses attached.

Round 2
Reviewer 3 Report
Yes, the new version resulted more clear and richer in details of all the experiment. You have profitably followed all the reviewers comments. Congratulations.